# Seroprevalence of anti-SARS-CoV-2 antibodies in women attending antenatal care in eastern Ethiopia: a facility-based surveillance

Nega Assefa ,[1,2] Lemma Demissie Regassa ,[1] Zelalem Teklemariam,[1] Joseph Oundo,[2] Lola Madrid,[2] Yadeta Dessie,[1] JAG Scott[2]

¹College of Health and Medical Sciences, Haramaya University, Harar, Ethiopia
²London School of Hygiene & Tropical Medicine, London, UK

**Correspondence to**
Dr Nega Assefa;
negaassefa@yahoo.com

## ABSTRACT

**Objective** We conducted serosurveillance of anti-SARS-CoV-2 antibodies among pregnant women attending their first antenatal care.

**Setting** The surveillance was set in one referral hospital in Harar, one district hospital and one health centre located in Haramaya district in rural eastern Ethiopia.

**Participants** We collected questionnaire data and a blood sample from 3312 pregnant women between 1 April 2020 and 31 March 2021. We selected 1447 blood samples at random and assayed these for anti-SARS-CoV-2 antibodies at Hararghe Health Research laboratory using WANTAI SARS-CoV-2 Rapid Test for total immunoglobulin.

**Outcome** We assayed for anti-SARS-CoV-2 antibodies and temporal trends in seroprevalence were analysed with a $\chi^2$ test for trend and multivariable binomial regression.

**Results** Among 1447 sera tested, 83 were positive for anti-SARS-CoV-2 antibodies giving a crude seroprevalence of 5.7% (95% CI 4.6% to 7.0%). Of 160 samples tested in April–May 2020, none was seropositive; the first seropositive sample was identified in June and seroprevalence rose steadily thereafter ($\chi^2$ test for trend, p=0.003) reaching a peak of 11.8% in February 2021. In the multivariable model, seroprevalence was approximately 3% higher in first-trimester mothers compared with later presentations, and rose by 0.75% (95% CI 0.31% to 1.20%) per month of calendar time.

**Conclusions** This clinical convenience sample illustrates the dynamic of the SARS-CoV-2 epidemic in pregnant women in eastern Ethiopia; infection was rare before June 2020 but it spread in a linear fashion thereafter, rather than following intermittent waves, and reached 10% by the beginning of 2021. After 1 year of surveillance, most pregnant mothers remained susceptible.

## INTRODUCTION

In Ethiopia, the first case of COVID-19 was reported on 13 March 2020. By the end of March 2021, there were 206 589 reports of COVID-19 infection and 2865 COVID-19-related deaths. In a country with an estimated population, in 2019, of 112 million this represents a cumulative incidence of SARS-CoV-2 infection of only 0.2% after a full year

### Strengths and limitations of this study

► The surveillance was initiated quickly at the start of the pandemic and was pursued with consistent methods over a full calendar year.

► Pregnant women are consistently available for surveillance throughout movement restrictions providing a practical and valid survey of seroprevalence trends.

► Results from pregnant women may not be fully representative of older or younger women, nor of men at any age.

► SARS-CoV-2 antibodies were assayed using a lateral flow device which, though convenient, has inferior performance characteristics to ELISA.

of transmission. Many cases of COVID-19 present with mild symptoms and, in Ethiopia, three quarters of PCR-positive cases have no symptoms.[1 2] Access to PCR testing in Ethiopia is also sparse. Monitoring the epidemic by detecting symptomatic cases is, therefore, highly insensitive. In these circumstances, seroprevalence of anti-SARS-CoV-2 antibodies can provide a more accurate estimator of cumulative incidence. Undertaking community sero-sampling during the pandemic is difficult when travel and household access are constrained by control measures. Expectant mothers, however, are likely to continue to seek health services throughout the pandemic and they can be used as a continuously available proxy population to estimate the cumulative incidence among young adults.[3–5] In addition, serological surveillance is simple to implement at antenatal clinic (ANC) visits because anti-SARS-CoV-2 antibodies can be assayed in the residual blood volumes of routine samples collected for clinical screening for anaemia and maternal infectious diseases.

Planning and provision of healthcare during a major epidemic like COVID-19 pose substantial logistical and clinical challenges. Information on the shape of the epidemic curve is critical to inform public health responses. The dynamics of seroprevalence reflect the epidemic curve and can provide an estimate of the effective reproduction number. Seroprevalence also indicates the likelihood of approaching transmission control through population immunity. This study aimed to assess the trend in seroprevalence of anti-SARS-CoV-2 antibodies throughout the first year of the epidemic by assaying anti-SARS-CoV-2 antibodies among pregnant women attending ANC at three different health facilities in the area around Harar, eastern Ethiopia.

## MATERIAL AND METHODS
### Study area and period
The surveillance was conducted between 1 April 2020 and 31 March 2021 at Awoday Health Centre and Haramaya District Hospital, both in Haramaya district(rural), and in Hiwot Fana Specialized Referral University Hospital in Harar(urban). Hiwot Fana is the largest referral and teaching hospital in eastern Ethiopia and receives tertiary referrals from Harari region, East Oromia, Somali region and Dire Dawa city. It is one of the 10 regional centres designated by the Federal Ministry of Health to manage the COVID-19 epidemic. Haramaya Hospital was rapidly designated a COVID-19 treatment facility and women seeking ANC services were therefore referred to Awoday Health Centre after 16 April 2020. Ethiopia began to roll out COVID-19 vaccine in the first quarter of 2021; however, no doses were given in the study area during the period this analysis covers.

### Study design, population and sample size
At the end of March 2020, we integrated health facility-based surveillance into the routine clinical care of pregnant women at Hiwot Fana Hospital, Awoday Health Centre and Haramaya Hospital. The study population comprised 3390 pregnant women attending their first antenatal care in these three facilities during the surveillance period. Because we had fewer test kits available than there were samples available from the clinic, we selected a random sample for analysis stratified on month. Initially, we decided to select 144 samples per month; as the seroprevalence was very low in the first 3 months we reset the sample size to 80 per month from July onwards and increased it to 160 per month in December once the seroprevalence reached 5%. In October, the number of samples collected was lower than the desired sample size and we therefore tested all of the samples (see online supplemental table).

A total of 78 women were excluded because they were not willing to provide a blood sample. Routine antenatal care includes serological screening for HIV, syphilis and toxoplasma infection during pregnancy undertaken in two blood samples; the first blood sample is taken at 16 weeks' gestation or at the first ANC visit, if later.

Sociodemographic data and information on pregnancy, clinical symptoms of COVID-19 and comorbidities were collected by trained nurses. COVID-19 symptoms were defined as at least one of cough, fever, headache or difficulty breathing. Data quality and completeness were checked daily.

### Laboratory analyses
For the anti-SARS-CoV-2 antibodies test, residual blood samples from the routine ANC tests were transferred to a test-tube containing clot activator by trained medical laboratory technologists working in each health facility. The blood samples were allowed to clot and serum was separated by centrifugation at 3000 RPM for 10 min. Serum samples were stored at 2°C–8°C at each site and transported in cool boxes to Hararghe Health Research Laboratory where they were stored at −80°C.

Samples were tested using WANTAI SARS-CoV-2 Ab Rapid Test. The test is a lateral flow assay in a cassette format designed for the qualitative detection of total antibodies to SARS-CoV-2 in human serum. The receptor-binding domain of the SARS-CoV-2 spike protein is bound at the Test Zone (T) and antibodies are bound at the Control Zone (C) of the cassette. The test has a sensitivity of 100% and specificity of 98.8% under validation performed by the manufacturer[6]; independent validation of the test found a sensitivity of 89%.[7] All the stored serum samples, tests reagents and cassettes were brought to room temperature (15°C–30°C) 30 min before performing the test and checked for defects. Then, a 10 µL of serum specimen and two drops of diluent buffer were added into the specimen window. Results were read and interpreted as reactive/positive (red line on C and T) or non-reactive/negative (red line on C) after 15–20 min according to manufacturer's instruction.[6 7] Serum samples were taken ≥14 days after a positive PCR test from COVID-19 infected individuals were used as positive control. Samples were tested in batches of 50–60 by a single operator. Assays without a valid reaction on the control line were rejected and the assay was repeated on a new kit.

### Patient and public involvement statement
Because the surveillance was set up urgently at the beginning of the pandemic we were not able to involve participants or the public in the design or set-up and because it was designed as an anonymous surveillance we were not able to provide individual feedback of the results to the participants. We have provided feedback of these high-level results through our existing community engagement exercises, including local radio programmes, meetings with local leaders and communication through health workers for onward dissemination.

### Statistical analysis
We used STATA V.16.0 for statistical management and analysis. We selected a random sample of participants

each month using the runiform function. We estimated unadjusted seroprevalence of SARS-CoV-2 IgG antibody with a 95% CI. We did not make adjustment for the test performance characteristics because the manufacturer's validation assay found very high sensitivity and specificity. We examined the univariate association between individual characteristics and seropositivity using $\chi^2$ test and multivariable associations using binomial regression. The trend in seropositivity with time was tested with a $\chi^2$ test for trend and in the multivariable model. Data used in the analysis are available online (dataverse.harvard.edu).[8]

### Ethical consideration

The surveillance was confined to residual clinical blood sample testing and anonymised data were collected using checklists to extract data from ANC cards. Bar codes, representing an anonymous unique identity number, were used to link extracted clinical and demographic data with test sample results. The exercise was conducted as part of a public health surveillance, with the approval of the directors of each of the three health facilities, and the results were made available to health facilities, the Regional Health Bureaux(Harari and Oromia) and the Ethiopian Public Health Institute. All ANC attendees were informed that the clinic was participating in an anonymous surveillance and mothers were made aware that the residual volumes of their blood samples would be made available to the surveillance laboratory. Written individual informed consent was not obtained.

## RESULTS

### Demographic characteristics of the study participants

Between 1 April 2020 and 31 March 2021 there were 3390 first visits to the ANCs; 1568 (46.24%) at Hiwot Fana Hospital, 1823 (53.75%) at Awoday Health Centre and Haramaya Hospital. At these, we interviewed and collected blood samples from 3312 women. We tested a random sample of 1447 blood specimens (table 1); 752 (52%) were from Haramaya district (Awoday Health Centre and Haramaya Hospital) and 695 (48%) were from Hiwot Fana Hospital; 984 (68%) were urban residents.

Among the population sample tested, the mean (SD) age was 23.9 (4.7) years and ages ranged from 15 to 45 years. The mean (SD) number of children per mother was 1.5 (1.8). The median (IQR) gestational age at the first antenatal visit was 20 (13–28) weeks. Only 51 (3.5%)

**Table 1** Characteristics of 1447 pregnant mothers attending their first antenatal care at the study clinics between April 2020 and March 2021 and sampled at random for the study

| Characteristics | Haramaya District | | Hiwot Fana Hospital | | Total | |
|---|---|---|---|---|---|---|
| | n | % | n | % | n | % |
| **Age in years** | | | | | | |
| 14–19 | 98 | 13.0 | 74 | 13.5 | 192 | 13.3 |
| 20–24 | 341 | 45.4 | 268 | 35.6 | 588 | 40.7 |
| 25–29 | 181 | 24.1 | 219 | 31.6 | 400 | 27.7 |
| 30–34 | 110 | 14.6 | 100 | 14.4 | 210 | 14.5 |
| ≥35 | 22 | 2.9 | 35 | 4.9 | 56 | 3.9 |
| **Residence** | | | | | | |
| Urban | 577 | 76.8 | 407 | 58.7 | 984 | 68.1 |
| Rural | 174 | 23.2 | 287 | 41.4 | 461 | 31.9 |
| **Number of children** | | | | | | |
| None | 298 | 40.0 | 254 | 36.8 | 552 | 38.4 |
| 1–5 | 422 | 56.6 | 401 | 58.1 | 823 | 57.3 |
| 6–10 | 26 | 3.5 | 35 | 5.1 | 61 | 4.3 |
| **Trimester of visit** | | | | | | |
| First | 223 | 29.7 | 143 | 20.6 | 366 | 25.3 |
| Second | 416 | 55.3 | 242 | 34.8 | 658 | 45.5 |
| Third | 113 | 15.0 | 310 | 44.6 | 423 | 29.2 |
| **Comorbidities** | | | | | | |
| None | 750 | 99.7 | 689 | 99.1 | 1439 | 99.5 |
| At least one* | 2 | 0.3 | 6 | 0.9 | 8 | 0.6 |
| **COVID-19 symptoms†** | | | | | | |
| No | 721 | 96.4 | 668 | 96.5 | 1389 | 96.5 |
| Yes | 27 | 3.6 | 24 | 3.5 | 51 | 3.5 |

*Chronic liver, renal, cardiovascular or 'other' disease.
†At least one of cough, fever, headache or difficulty breathing.

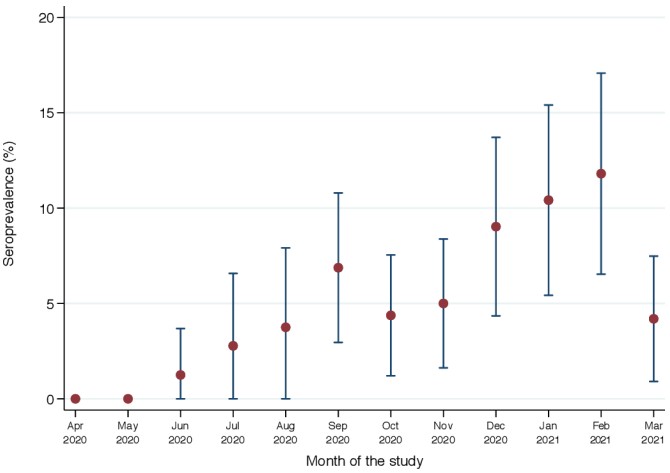

**Figure 1** Temporal trend of seroprevalence of anti-SARS-CoV-2 antibodies among pregnant women presenting for first antenatal care in three antenatal clinic facilities, eastern Ethiopia, between 1 April 2020 and 31 March 2021.

had COVID-19 symptoms at the time of sampling and 8 (<1%) had a history of comorbidity, given as chronic liver, renal, cardiovascular or 'other' disease. Respiratory diseases, chronic neurological disease, diabetes mellitus and cancer were not reported by any participant.

### Seroprevalence of SARS-CoV-2 antibodies

Of 1447 samples tested, 83 (5.7%, 95% CI 4.6 to 7.0%) were positive for anti-SARS-CoV-2 antibodies. The first seropositive sample was identified on 11 June 2020, and seroprevalence rose progressively thereafter, with the exception of March 2021, where it dropped sharply ($\chi^2$ for trend for the whole year, p=0.003; figure 1).

Seroprevalence also varied significantly by trimester of pregnancy and comorbidity but not by clinic, residence or COVID-19 symptoms (table 2). Given the linear growth in seroprevalence (figure 1) and better model fit based on Bayesian information criterion, we modelled prevalence associations as risk differences rather than risk ratios.

**Table 2** Seroprevalence of anti-SARS-CoV-2 antibodies by participant characteristics

| Characteristics | Tested N | Seropositive n | Seroprevalence % | $\chi^2$ test P value |
|---|---|---|---|---|
| Age in years | | | | |
| 14–19 | 192 | 7 | 3.7 | 0.19 |
| 20–24 | 588 | 42 | 7.1 | |
| 25–29 | 400 | 24 | 6.0 | |
| 30–34 | 210 | 9 | 4.3 | |
| ≥35 | 56 | 1 | 1.8 | |
| ANC | | | | |
| Hiwot Fana Hospital | 695 | 35 | 5.0 | 0.260 |
| Haramaya Hospital | 19 | 0 | 0.0 | |
| Awoday Health Centre | 733 | 48 | 6.6 | |
| Residence | | | | |
| Urban | 984 | 57 | 5.8 | 0.910 |
| Rural | 461 | 26 | 5.6 | |
| Number of children | | | | |
| None | 520 | 32 | 5.8 | 0.370 |
| 1–5 | 778 | 45 | 5.5 | |
| 6–10 | 55 | 6 | 9.8 | |
| Trimester of visit | | | | |
| First | 366 | 31 | 8.5 | 0.034 |
| Second | 658 | 32 | 4.9 | |
| Third | 423 | 20 | 4.8 | |
| Comorbidities | | | | |
| None | 1439 | 81 | 5.6 | 0.019 |
| At least one* | 8 | 2 | 25.0 | |
| COVID symptoms† | | | | |
| No | 1389 | 82 | 5.9 | 0.240 |
| Yes | 51 | 1 | 2.0 | |

*Chronic liver, renal, cardiovascular or 'other' disease.
†At least one of cough, fever, headache or difficulty breathing.
ANC, antenatal clinic.

In a multivariable binomial regression model, the prevalence difference was −3.2% (95% CI −6.7% to −0.4%) and −3.0% (95% CI −6.8% to −0.8%) among women in their second and third trimesters, respectively, compared with those in the first trimester and the prevalence difference was 0.75% (95% CI 0.31% to 1.20%) per month of calendar time.

## DISCUSSION

The study provides a simple description of the dynamic of SARS-CoV-2 epidemic in an area where reliable data are extremely rare. In a population of attendees at ANCs in three sites in eastern Ethiopia, antibodies against SARS-CoV-2 first appeared in June 2020 and seroprevalence rose steadily month on month reaching approximately 10% at the beginning of 2021. Although the point estimate for March 2021 is substantially lower, the data as a whole evince a strong linear trend and this single estimate is most likely to have deviated from the general direction by chance. If these results are reliable, they indicate that the epidemic is progressing here at a considerably lower rate than in other settings in East Africa and that the greater majority of the population remains uninfected, suggesting that the epidemic is still at an early stage.

The principal limitations of the study are the potential generalisability of the population under surveillance and the validity of the serological assay employed. Pregnant women have been used as an indicator population in prior pandemics, including HIV,[9] but also for SARS-CoV-2, both in high-income settings[4 5 10–14] and low-income and middle-income settings, including in neighbouring Kenya.[3] The principal advantage of sampling pregnant women is that they remain one of the few patient groups for whom health services cannot be postponed until after the pandemic has passed. They are permitted and encouraged to attend even in the face of social and movement restrictions, and so provide a consistent and reliable sampling group. The principal limitation of this group is their restriction on age and sex, however, in most settings, including other East African countries, seroprevalence does not vary significantly by sex and the cumulative incidence in women is likely to represent the infection history of both sexes.[15–20] Similarly, in most settings young adults are the group most likely to be infected by SARS-CoV-2 and so the seroprevalence estimates here are likely to represent the highest risk in the whole population; other age groups, particularly children and the elderly, are likely to have lower seroprevalence.[17 21]

The WHO has deprecated the use of rapid tests for SARS-CoV-2 antibodies for individual diagnosis but recognises their potential value in research.[22] WHO has also recommended and endorsed quantitative analysis of IgG antibodies using ELISA and has distributed the WANTAI ELISA kit to countries undertaking serosurveillance. Reliance on ELISA, however, limits the range of settings in which serosurveillance can be undertaken and lateral flow tests have been successfully employed for recurrent community-based nationwide surveys in the UK.[23] When seroprevalence is low,

as at the beginning of our study, an assay with imperfect specificity may detect more false positives than true positives. The specificity of the WANTAI rapid test has been estimated by the manufacturer at 98.8%. We assayed 80 samples each month in April and May 2020 without observing a single positive test, suggesting that the specificity is indeed very high. Even if the positive results identified in June included false positives, the progressive rise in seropositivity with time is most unlikely to be influenced materially by a small fraction of false positive results.

The assay sensitivity may also be imperfect in detecting prior infection because the assay was originally calibrated against sera from symptomatic cases, who generally have higher antibody levels than asymptomatic individuals,[24] and because pregnant women who were infected several months ago may have experienced waning of antibody levels and seroreversion.[25–29] In general, seroreversion is less problematic in assays that measure total immunoglobulin and in those that target spike antigens, compared with nucleocapsid antigens,[27 30] so the problem of waning in this study is unlikely to be substantial. Furthermore, if sensitivity is unlikely to decline over time, imperfect sensitivity would not affect the shape of the rising seroprevalence line, though it would underestimate the gradient. If, as estimated in one validation study, the WANTAI rapid test has a sensitivity of only 89%,[7] adjustment for test-performance characteristics would elevate our reported seroprevalence results by a factor of 1.12.

The results are in contrast to most other settings studied which record a sharp take-off in seroprevalence once transmission begins, often rising quickly to high levels. In Kenya for example, women attending an ANC in Kilifi had seroprevalence of 0%, 2% and 11% in consecutive months September–November 2021; those attending ANC in Nairobi had a seroprevalence of 50% in August 2020.[3] The pattern illustrated in eastern Ethiopia is more indicative of a gradually spreading epidemic curve suggesting an effective reproduction number much closer to 1.

In Juba, South Sudan, seroprevalence was 22% in a household survey in August–September 2020[20]; in Kenya, a national estimate for seroprevalence, based on testing blood transfusion donors, was 4.3% in May 2020 18% and 9.1% 2 months later.[15] Healthcare workers in Nairobi, Kenya, had a seroprevalence of 44% in August 2020; those in two rural hospitals had seroprevalence of 12%–13% in November 2020.[15] Finally, in Addis Ababa seroprevalence, estimated in May 2020, was 3.0%.[31] Although all these studies used different laboratory assays and varied statistical adjustments, collectively, they suggest that transmission in eastern Ethiopia began later than in much of the rest of the region, including the state capital, and has progressed more slowly.

## CONCLUSION

In summary, if seroprevalence is a reliable indicator of cumulative incidence, SARS-CoV-2 infection is spreading slowly but steadily in eastern Ethiopia. This contrasts sharply with the recurrent waves of PCR-positive infections apparent in

the national surveillance system. One year after the start of the epidemic, approximately 10% of women attending ANCs are seropositive implying that the COVID-19 epidemic is still at an early stage in eastern Ethiopia.

**Acknowledgements** Investigators would like to thank authorities of the health facilities for facilitating the surveillance and pregnant women for voluntarily participating in the study.

**Contributors** NA and JAGS lead overall surveillance. NA, JAGS, ZT and LDR develop concept of the surveillance, analysed the data and wrote the manuscript. ZT provided microbiome data analysis and interpretation. NA, JAGS, ZT, LDR, LM, JO and YD reviewed the manuscript and give critical feedback. All authors approved the submission of the manuscript to the journal. NA is the guarantor.

**Funding** Activities of the surveillance and Laboratory analysis was supported by Emory University as part of the CHAMPS network (www.champshealth.org). AS was funded by The Wellcome Trust through a Senior Research Fellowship (214320).

**Competing interests** None declared.

**Patient and public involvement** Patients and/or the public were involved in the design, or conduct, or reporting, or dissemination plans of this research. Refer to the Methods section for further details.

**Patient consent for publication** Consent obtained directly from patient(s)

**Ethics approval** The surveillance was approved by the Institutional Health Research Ethical Review Committee of the College of Health and Medical Sciences, Haramaya University, Ethiopia with clearance number 123/2021.

**Provenance and peer review** Not commissioned; externally peer reviewed.

**Data availability statement** Data are available in a public, open access repository. [Dataset] [8] Assefa N, Demissie L, Teklemarriam Z, Oundo J, Madrid L, Dessie Y, et al. *Data from*: Seroprevalence of anti–SARS-CoV-2 antibodies in women attending antenatal care in eastern Ethiopia: V1 ed: Harvard Dataverse; June 9, 2021. doi:10.7910/DVN/XIWCXN.

**ORCID iDs**
Nega Assefa http://orcid.org/0000-0003-0341-2329
Lemma Demissie Regassa http://orcid.org/0000-0002-5461-5348

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
