## [Reviewer comments · BMJ Open]

ARTICLE DETAILS

TITLE (PROVISIONAL)	Seroprevalence of anti-SARS-CoV-2 antibodies in women attending antenatal care in eastern Ethiopia: a facility-based surveillance
AUTHORS	Assefa, Nega; Regassa, Lemma; Teklemariam, Zelalem; Oundo, Joseph; Madrid-Castillo, Lola; Dessie, Yadeta; Scott, A

VERSION 1 – REVIEW

REVIEWER	Tonry, Claire
REVIEW RETURNED	13-Aug-2021

GENERAL COMMENTS	Overall Impression The authors present a large-scale study on the seroprevalence of SARS-CoV-2 infection in the Ethiopian population between March 2020 and March 2021. However, they have only monitored seroprevalence in pregnant women, under the age of 45. Although the authors do recognize that this is a limitation to their study, they still claim (inappropriately) that their findings “illustrates the dynamic of the SARS-CoV-2 epidemic in young adults in eastern Ethiopia”. I do not believe that this conclusion can be made based on results from a very niche subset of the Ethiopian population. Aside from the fact that their findings are limited by gender and age bias, it is not known yet what effect pregnancy has on susceptibility to SARS-CoV-2 infection. Pregnancy is known to alter the immune system. Therefore, these women may have a slight immune advantage compared to non-pregnant women and this may account for why incidence of SAR-CoV-2 was so low. If such considerations were made during the design of the study, a more convincing argument must be made for why this population was considered to be reflective of the general young population. There also needs to be some discussion on the potential impact of pregnancy on SARS-CoV-2 immunity, backed up by any evidence that is available. Overall, the study is underwhelming and it is not clear how these results will have any meaningful impact on clinical management of Covid-19 or what new information can be gleaned from this study. I can appreciate the difficulty in getting access to ‘healthy people’ for asymptomatic testing of SARS-CoV-2 infection under controlled (hospital setting) circumstances. However, if pregnant women are the only available study participants, I feel that the study should have been designed to address a more pertinent and relevant clinical question. For example, the authors state that “The effect of COVID infection on birth outcomes needs to be investigated”. Perhaps if they had made this the primary aim of this study, it would have contributed something to the field of knowledge. However, pregnancy outcomes in this population are not reported here, which seems to be a missed opportunity. Major Comments:
--

In addition to the observations made above, there are some weaknesses associated with this study that could be addressed:

1. A lateral flow test was used to measure SARS-CoV-2 antibody levels. Lateral flow tests are known to be much less sensitive than ELISA and PCR and the sensitivity and specificity of lateral flow tests can vary greatly between suppliers (this is outlined in REF18). Although authors claim that the test used has sensitivity of 100% and specificity of 98.8% this is based solely on the manufacturers data. In fact, these values appear to be suspiciously high for a lateral flow test. Authors should include data from their own in-house verification of assay performance and potentially adjust their findings based on these test performance characteristics. Authors should also provide details on what quality control measures were implemented to ensure that there were no batch-to-batch variations throughout the 12 months of the study. Was the analysis of all 1,000+ samples conducted by the same operator/in the same lab? Given that this analysis was being performed with the intention of publication and adding knowledge to the field, samples (or at least some samples) should have been tested in duplicate to validate results.
2. Patient Demographics data only emphasizes the fact that women included in the study cannot be used as representative subjects for the general population. Authors have not reported on other potentially relevant socioeconomic factors such as occupation and marital status, number of people in the home, occupation of spouse etc. Some of these factors would likely influence potential exposure of these women to SARS-CoV2 infection. Authors could also include a table comparing demographic and clinical history/symptomatic data collected between women who tested positive and (matched) women who tested negative for SARS-CoV-2 (matched based on age, location etc.)
3. Only one third of women who attended ANC were tested. Authors have not provided sufficient detail on their selection process. Were the same number/proportion of women selected from each time point? Was the selection balanced based on location, age range and gestational stage? If blood samples were obtained for 3306 study participants, is there any reason that all were not tested over the course of the 12-month study? How have authors ensured that their selection would be representative of the full cohort?
4. The manuscript does not include follow-up data on any of the patients enrolled in the study. Pregnancy outcome data would have been important to include, especially considering observations that women in the early stages of pregnancy were slightly more likely to test positive for SARS-CoV-2 than women in later stages. It would also have been of interest to learn if antibody levels are maintained in women who tested positive over time or if any women became ill following a positive result. The lack of follow up is a significant limitation to this study
5. The main observation reported from this work is a steady upward trend of seropositive cases over the course of the year. Although authors believe this to be a relatively unique observation, given that many other African regions observed more dramatic spikes in incidences, it seems doubtful that this trend is truly reflective of what was happening in the general population. Authors could perhaps strengthen their discussion of these results if provided in context of how the pandemic was being managed by Ethiopian government compared to other African regions i.e. were there more restrictions in place etc.?

	6. Authors should provide details on how many women were tested each month. It would have been interesting to monitor the presence of antibodies longitudinally in this population and report on incidences of seroconversion. Another advantage to using an ELISA test as opposed to the Lateral flow is that antibody titres could have been analysed. It is not recorded anywhere whether any of the women who tested positive became ill or had been ill prior to their hospital visit. Minor Comments Throughout the manuscript there are numerous typos and words missing from sentences e.g. in the Key Summaries section, point 1: "Information on the cumulative incidence of SARS-CoV-2 in is scarce....." The authors refer to the Covid-19 pandemic as an epidemic throughout the manuscript. This needs to be corrected. It would also be of interest to the reader to know if vaccines were available to these women during the course of this study? Did any women accept the vaccine if offered to them? What has vaccine uptake been like in Ethiopia? Authors should use 'Chi-Square test' instead of the x2 symbol when describing statistical analyses used
--	---

REVIEWER	Rysavy, Mary B The University of Iowa
REVIEW RETURNED	31-Aug-2021

GENERAL COMMENTS	This was an excellent study assessing the trend in seroprevalence of COVID-19 antibodies in a population of young women presenting for antenatal care in eastern Ethiopia. The authors have a very clearly written paper. They have a clear and simple objective. The methods are described appropriately. Their strengths and limitations are adequately discussed and seem reasonable. I agree that this was an excellent population to sample, as pregnant women are unique in their need to continue to seek regular healthcare, despite the pandemic concerns. I would recommend that the authors revise the "key summaries" section for appropriate English grammar. Lines 54, 58, and 61 all contain errors in English language grammar. The rest of the paper is overall appropriate, however, I again noted English grammar errors in lines 136 and 144. With these minor revisions, I think the paper is suitable for publication.
---

REVIEWER	Tartari, Ermira University of Malta, Faculty of Health Sciences
REVIEW RETURNED	03-Sep-2021

GENERAL COMMENTS	This is an interesting study to assess the seroprevalence trend of anti-SARS-CoV-2 antibodies, and importantly providing data and information from eastern Ethiopia which is scarce on the topic. Some points for the authors to consider: Key summaries (page 5)  - Line 54 "SARS-CoV...did you mean "infection"? Please add - Line 58 "a healthy looking..What? - Line 61 is not clear Methodology
---

	- Can you clarify further the recruitment procedure? did participants provide informed consent? Were they informed about voluntary participation to the study? - You mention "anonymised questionnaire" " we collected questionnaire data" Can you clarify what data was collected as this is not clear? is the questionnaire available? Please review again the "Patient and public involvement section" some language errors - Line 132 "...much attention..has..."
--	---

VERSION 1 – AUTHOR RESPONSE

Reviewers Comments

Reviewer 1.

1. The authors present a large-scale study on the seroprevalence of SARS-CoV-2 infection in the Ethiopian population between March 2020 and March 2021. However, they have only monitored seroprevalence in pregnant women, under the age of 45. Although the authors do recognize that this is a limitation to their study, they still claim (inappropriately) that their findings “illustrates the dynamic of the SARS-CoV-2 epidemic in young adults in eastern Ethiopia”. I do not believe that this conclusion can be made based on results from a very niche subset of the Ethiopian population. Aside from the fact that their findings are limited by gender and age bias, it is not known yet what effect pregnancy has on susceptibility to SARS-CoV-2 infection. Pregnancy is known to alter the immune system. Therefore, these women may have a slight immune advantage compared to non-pregnant women and this may account for why incidence of SAR-CoV-2 was so low. If such considerations were made during the design of the study, a more convincing argument must be made for why this population was considered to be reflective of the general young population. There also needs to be some discussion on the potential impact of pregnancy on SARS-CoV-2 immunity, backed up by any evidence that is available.

The impact of pregnancy on susceptibility to COVID-19 is unclear (Wastnedge EAN: doi: 10.1152/physrev.00024.2020). However, if we accept that pregnant mothers do not change in their susceptibility to COVID-19 infection over calendar time (pregnant mothers in early 2021 were just as susceptible to infection, given exposure, as pregnant mothers in mid 2020) then regardless of any potential effect of pregnancy on susceptibility the study remains valid in illustrating the dynamic of seroprevalence, that is, the take-off from zero seroprevalence and the nature of the rising seroprevalence.

In the short term, before waning becomes apparent, seroprevalence is a marker of cumulative incidence. This helps public health planners to see where the infection is spreading and when. Pregnant mothers may not be representative of all young women and are not necessarily representative of young men with regard to COVID-19 infection; nonetheless, pregnant mothers have been widely used in the evaluation of the dynamic of local COVID epidemics (Francesca C, et.al 2021: doi.org/10.1101/2020.06.17.20134098; Mattern J & et.al, 202 doi.org/10.1371/journal.pone.0240782) and seroprevalence studies of SARS-CoV-2 both regionally (Adetifa, I. M. O. et al, 10.1101/2021.02.09.21251404v1 (2021); Wiens, K. E. et al., doi:10.1101/2021.03.08.21253009 (2021)) and further afield (Pollan, M. et al; doi:10.1016/S0140-6736(20)31483-5 (2020)) observe little difference in the seroprevalence with sex, indicating that a single sex sample remains useful.

The conclusion referred to is: “This clinical convenience sample illustrates the dynamic of the SARS-CoV-2 epidemic in young adults in eastern Ethiopia; infection was rare before June 2020 but it spread in a linear fashion thereafter, rather than following intermittent waves, and reached 10% by the beginning of 2021” The study is an illustration of the dynamic, it is in young adults, it is in two centres in Eastern Ethiopia. The magnitude of the final figure of 10% may not be representative of all women in this area but the conclusion is careful to delineate that this estimate refers to a clinical convenience sample. We disagree with the impressions of the reviewer and affirm that the conclusion stays within the limits of the data presented.

2. Overall, the study is underwhelming and it is not clear how these results will have any meaningful impact on clinical management of Covid-19 or what new information can be gleaned from this study.

We didn't do the study to overwhelm but to inform. Ethiopia is a country with a population of 112 million people and prior to our submission there was only one SARS-CoV-2 antibody serosurvey reported – and that was restricted to the capital city. Access to PCR testing has been extremely limited and so the Federal Ministry of Health have only a limited picture of the timing and extent of the SARS-CoV-2 epidemic across the country. This serosurvey therefore provides a substantial increase in the amount of information available in country. This does not make it remarkable but it does make it useful for public health planning. There was never any intention of using the results for clinical management; the study was a public health surveillance exercise as defined in the objectives (Introduction) “This study aimed to assess the trend in seroprevalence of anti-SARS-CoV-2 antibodies throughout the first year of the epidemic”

3. I can appreciate the difficulty in getting access to ‘healthy people’ for asymptomatic testing of SARS-CoV-2 infection under controlled (hospital setting) circumstances. However, if pregnant women are the only available study participants, I feel that the study should have been designed to address a more pertinent and relevant clinical question. For example, the authors state that “The effect of COVID infection on birth outcomes needs to be investigated”. Perhaps if they had made this the primary aim of this study, it would have contributed something to the field of knowledge. However, pregnancy outcomes in this population are not reported here, which seems to be a missed opportunity.

At the very beginning of the pandemic we realised that there would be severe constraints on test availability in this relatively poor part of Ethiopia. To monitor the epidemic at a public health level, we looked around rapidly for a population that would be available throughout lockdowns, would remain consistent over time, and would provide a measure of rising seroprevalence to indicate when SARS-CoV-2 arrived and started infecting the population. Pregnant women attending ANC met these criteria both in theory and (as evidenced from the report) in practice. We never intended to study the impact of SARS-CoV-2 infection on pregnancy outcomes. The ‘field of knowledge’ to which we hoped to contribute was the timing of the epidemic of SARS-CoV-2 in Eastern Ethiopia.

Major Comments:

In addition to the observations made above, there are some weaknesses associated with this study that could be addressed:

1. A lateral flow test was used to measure SARS-CoV-2 antibody levels. Lateral flow tests are known to be much less sensitive than ELISA and PCR and the sensitivity and specificity of lateral flow tests can vary greatly between suppliers (this is outlined in REF18). Although authors claim that the test used has sensitivity of 100% and specificity of 98.8% this is based solely on the manufacturers data. In fact, these values appear to be suspiciously high for a lateral flow test. Authors should include data from their own in-house verification of assay performance and potentially adjust their findings based on these test performance

characteristics. Authors should also provide details on what quality control measures were implemented to ensure that there were no batch-to-batch variations throughout the 12 months of the study. Was the analysis of all 1,000+ samples conducted by the same operator/in the same lab? Given that this analysis was being performed with the intention of publication and adding knowledge to the field, samples (or at least some samples) should have been tested in duplicate to validate results.

We have elaborated on the laboratory practices for the lateral flow test in the methods and included a note on the number of batches and operators and the quality control application and results. Given the study was conducted in a very low-resource setting, we decided to invest our limited resources on single (not duplicate testing) of a lateral flow test from a reputable manufacturer (their ELISA test is the sole WHO-approved assay for LMICs) and rely upon the validations of others. This includes one validation by the manufacturer and another by an independent assessor. We are very clear about the evidence: “The test has a sensitivity of 100% and specificity of 98.8% under validation performed by the manufacturer; independent validation of the test found a sensitivity of 89% [ref 6]” Of note, the assumption of 100% sensitivity is conservative – as we noted in the discussion: “If, as estimated in one validation study, the WANTAI rapid test has a sensitivity of only 89% [6], adjustment for test-performance characteristics would elevate our reported seroprevalence results by a factor of 1.12.”

2. Patient Demographics data only emphasizes the fact that women included in the study cannot be used as representative subjects for the general population. Authors have not reported on other potentially relevant socioeconomic factors such as occupation and marital status, number of people in the home, occupation of spouse etc. Some of these factors would likely influence potential exposure of these women to SARS-CoV2 infection. Authors could also include a table comparing demographic and clinical history/symptomatic data collected between women who tested positive and (matched) women who tested negative for SARS-CoV-2 (matched based on age, location etc.)

Given the routine and anonymous nature of this surveillance, it is not possible to administer a wide-ranging questionnaire. The principal finding (as intended) is the timing of the arrival of infections in Eastern Ethiopia and the speed and dynamic of rising seroprevalence. We have assumed that socioeconomic factors remained constant over time and therefore are not significant confounders of this dynamic. Regarding a comparison of the demographic and symptomatic data collected between women who tested positive and women who tested negative for SARS-CoV-2 antibodies - this can be found in Table 2.

3. Only one third of women who attended ANC were tested. Authors have not provided sufficient detail on their selection process. Were the same number/proportion of women selected from each time point? Was the selection balanced based on location, age range and gestational stage? If blood samples were obtained for 3306 study participants, is there any reason that all were not tested over the course of the 12-month study? How have authors ensured that their selection would be representative of the full cohort?

The sampling is described briefly, and inappropriately, in the first paragraph of the results. We have moved it to the methods and expanded to explain the constraints on the random process. The total number tested was determined by the resources available to purchase lateral flow test kits. We have also provided details of the sample compared to the full cohort in an additional supplementary table to allow readers to see the distribution of the sample across demographic variables and across time.

4. The manuscript does not include follow-up data on any of the patients enrolled in the study. Pregnancy outcome data would have been important to include, especially considering

observations that women in the early stages of pregnancy were slightly more likely to test positive for SARS-CoV-2 than women in later stages. It would also have been of interest to learn if antibody levels are maintained in women who tested positive over time or if any women became ill following a positive result. The lack of follow up is a significant limitation to this study

Individual follow up is precluded in an anonymous public health surveillance. Nonetheless, the lack of individual follow up did not prevent us from achieving the objectives identified at the outset.

5. The main observation reported from this work is a steady upward trend of seropositive cases over the course of the year. Although authors believe this to be a relatively unique observation, given that many other African regions observed more dramatic spikes in incidences, it seems doubtful that this trend is truly reflective of what was happening in the general population. Authors could perhaps strengthen their discussion of these results if provided in context of how the pandemic was being managed by Ethiopian government compared to other African regions i.e. were there more restrictions in place etc.?

We do not make the claim that the shape of rising seroprevalence is relatively unique [sic]. The observation we make is: "SARS-CoV-2 infection is spreading slowly but steadily in eastern Ethiopia. This contrasts sharply with the recurrent waves of PCR-positive infections apparent in the national surveillance system." The data from the national surveillance system is in the public domain and easily accessible. A snapshot relating to the study period is captured below from the 'Our World in Data' website. Given the limited sample denominator in each month, reflected by broad confidence limits in our Figure1, it is inappropriate to compare month-by-month figures with other varying factors. Nonetheless, our Figure 1 does evince a linear growth trend and, without wanting to make too much of it, we have drawn the readers eye to this.

6. Authors should provide details on how many women were tested each month. It would have been interesting to monitor the presence of antibodies longitudinally in this population and report on incidences of seroconversion. Another advantage to using an ELISA test as

opposed to the Lateral flow is that antibody titres could have been analysed. It is not recorded anywhere whether any of the women who tested positive became ill or had been ill prior to their hospital visit.

Again, in the nature of anonymous public health surveillance, it is not possible to follow participants up longitudinally. We have included the number of women tested each month in the new supplementary table.

Minor Comments

1. Throughout the manuscript there are numerous typos and words missing from sentences e.g. in the Key Summaries section, point 1: "Information on the cumulative incidence of SARS-CoV-2 in is scarce....."

We have excised the Key Summaries section at the request of the editor.

2. The authors refer to the Covid-19 pandemic as an epidemic throughout the manuscript. This needs to be corrected.

We used the term pandemic (four times) when referring to the spread of SARS-CoV-2 across the globe and used the term epidemic (and 'epidemic curve') when referring to the spread of SARS-CoV-2 locally or regionally. Our objective is to help illustrate the epidemic curve in eastern Ethiopia (not the pandemic curve).

3. It would also be of interest to the reader to know if vaccines were available to these women during the course of this study? Did any women accept the vaccine if offered to them? What has vaccine uptake been like in Ethiopia?

Although COVID-19 vaccine was introduced into Ethiopia in the first quarter of 2021, no doses were supplied to our study area during the period of this surveillance. We have added a line in the methods to supply this information on page 5 line number 90-99.

4. Authors should use 'Chi-Square test' instead of the χ^2 symbol when describing statistical analyses used

We have replaced the Greek symbol for Chi with "Chi-square test" throughout.

Reviewer 2

1 I would recommend that the authors revise the "key summaries" section for appropriate English grammar. Lines 54, 58, and 61 all contain errors in English language grammar. The rest of the paper is overall appropriate, however, I again noted English grammar errors in lines 136 and 144. With these minor revisions, I think the paper is suitable for publication.

Thank you for the comment, the key summary has been revised, and only appropriate sections are retained Please see page 3 line 56-63.

Reviewer 3

- 1 Key summaries (page 5)
 - Line 54 "SARS-CoV...did you mean "infection"? Please add
 - Line 58 "a healthy looking..What?
 - Line 61 is not clear

Thank you for the comment, the key summary has been revised, and only appropriate sections are retained Please see page 3 line 56-63.

2 Methodology

Can you clarify further the recruitment procedure? did participants provide informed consent? Were they informed about voluntary participation to the study?

Attendees at the ANC clinics were informed about the surveillance and the use of residual blood sampling before the samples were collected as they provide blood for other routine tests. Written informed consent was not obtained. Please see details in Ethical consideration section of the method page 8 line numbers 155-167.

3 Methodology

- You mention "anonymised questionnaire" " we collected questionnaire data" Can you clarify what data was collected as this is not clear? is the questionnaire available?

This was a misleading contraction and it has been corrected, 'questionnaire data' has been changed to data were extracted from ANC cards using checklists. Please see details in Ethical consideration section of the method page 8 line numbers 155-167

4 Please review again the "Patient and public involvement section" some language errors
- Line 132 "...much attention has..."

This has been reviewed and modified. Please see section page 7 line numbers 139-145.

VERSION 2 – REVIEW

REVIEWER	Tonry, Claire
REVIEW RETURNED	13-Oct-2021

GENERAL COMMENTS	I apologise to the authors if my previous comments were overly-critical and mis-informed. I appreciate that authors have addressed the relevant major and minor comments, namely around providing details of the lateral flow testing and random sampling. I also feel that the inclusion of further discussion on the strengths and weaknesses of the study cohort with added references has helped with making the objectives and potential impact of this study very clear. The manuscript reads well and is suitable for publication with some very minor changes:  1. Introduction – ANC abbreviation used without prior description of that this abbreviation stands for 2. Page 12, line 234 – 'mis-spelling of evidence'
---

REVIEWER	Tartari, Ermira University of Malta, Faculty of Health Sciences
REVIEW RETURNED	30-Oct-2021

GENERAL COMMENTS	All the points that have been raised in the previous revision have been taken into consideration.
---